# The Effects of Ergosta-7,9(11),22-trien-3β-ol from *Antrodia camphorata* on the Biochemical Profile and Exercise Performance of Mice

**DOI:** 10.3390/molecules24071225

**Published:** 2019-03-28

**Authors:** Yi-Ming Chen, Hsin-Ching Sung, Yueh-Hsiung Kuo, Yi-Ju Hsu, Chi-Chang Huang, Hsin-Li Liang

**Affiliations:** 1Health Technology College, Jilin Sport University, Changchun 130022, China; yimingjon@gmail.com or 1003@jlsu.edu.cn; 2Department of Anatomy, College of Medicine, Chang Gung University, Taoyuan 33301, Taiwan; hcs@mail.cgu.edu.tw; 3Aesthetic Medical Center, Department of Dermatology, Chang Gung Memorial Hospital, Taoyuan 33301, Taiwan; 4Department of Chinese Pharmaceutical Sciences and Chinese Medicine Resources, China Medical University, Taichung 40402, Taiwan; yhkuo@ntu.edu.tw; 5Department of Biotechnology, Asia University, Taichung 41354, Taiwan; 6Chinese Medicine Research Center, China Medical University, Taichung 40402, Taiwan; 7Graduate Institute of Sports Science, National Taiwan Sport University, Taoyuan City 33301, Taiwan; 1041302@ntsu.edu.tw (Y.-J.H.); john5523@ntsu.edu.tw (C.-C.H.); 8Department of Critical Care Medicine, Kaohsiung Veterans General Hospital, Kaohsiung 81362, Taiwan

**Keywords:** *Antrodia camphorata* (AC), Ergosta-7,9(11),22-trien-3β-ol, EK100, anti-fatigue, exercise performance

## Abstract

*Antrodia camphorata* (AC) is a rare and unique mushroom that is difficult to cultivate. Previous studies have demonstrated the bioactivity of the compound Ergosta-7,9(11),22-trien-3β-ol (EK100) from AC in submerged culture. The purpose of this study is to evaluate the potential beneficial effects of EK100 on fatigue and ergogenic functions following physiological challenge. Male ICR (Institute of Cancer Research) mice were randomly divided into three groups (*n* = 8 per group) and orally administered EK100 for six weeks at 0 (Vehicle), 10 (EK100-1X), and 20 (EK100-2X) mg/kg/day. The six-week Ek100 supplementation significantly increased grip strength (*p* = 0.0051) in trend analysis. Anti-fatigue activity was evaluated using 15-min. acute exercise testing and measuring the levels of serum lactate, ammonia, glucose, blood urea nitrogen (BUN), and creatine kinase (CK) after a 15-min. swimming exercise. Our results indicate that AC supplementation leads to a dose-dependent decrease in serum lactate, ammonia, BUN, and CK activity after exercise and significantly increases serum glucose and glycogen content in liver tissues. Biochemical and histopathological data demonstrated that long term daily administration of EK100 for over six weeks (subacute toxicity) was safe. EK100’s anti-fatigue properties appear to be through the preservation of energy storage, increasing blood glucose and liver glycogen content, and decreasing the serum levels of lactate, ammonia, BUN, and CK. EK100 could potentially be used to improve exercise physiological adaptation, promote health, and as a potential ergogenic aid in combination with different nutrient strategies.

## 1. Introduction

The fruiting body of *Antrodia camphorata* (AC) is well known in Taiwan as a traditional Chinese medicine. AC is endemic to Taiwan and only grows on the local, large, evergreen broad-leaved tree Cinnamomum kanehirae Hayata (Lauraceae) [1,2]. The fruiting body and cultured mycelia of AC are rare and difficult to cultivate, containing fatty acids, lignans, phenyl derivatives, sesquiterpenes, steroids, and triterpenoids [3]. One of the bioactive compounds from the filtrate of AC in submerged culture is antrosterol (Ergosta-7,9(11),22-trien-3β-ol), which is also known as EK100 [2].

Ergosta-7,9(11),22-trien-3β-ol (EK100) is a derivative of ergosterol and it is almost found exclusively in fungi. It is frequently used by environmental microbiologists as an indicator of living fungal biomass, based on the assumption that ergosterol is labile and therefore rapidly degraded after the death of fungi [4]. It is a poorly soluble plant sterol with anti-tumor and anti-angiogenic activities. A strong drug candidate, it has shown potential to induce cancer cell death with no side effects [5]. The protective effects of EK100 against carbon tetrachloride (CCl_4_)-induced liver damage in mice have previously been observed [6]. The antidiabetic and antihyperlipidemic effects of EK100 have been demonstrated in a previous study [2] with mice that were fed on a high fat diet (HFD). EK100 appears to have a therapeutic potential for treatment of type 2 diabetes that is associated with hyperlipidemia, by regulating the levels of Glucose transporter type 4 (GLUT4), Phosphoenolpyruvate carboxykinase (PEPCK), Glucose 6-phosphatase (G6Pase), Sterol regulatory element-binding transcription factor 1 (SREBP1c), Sterol regulatory element-binding protein 2 (SREBP2), Apolipoprotein A1 (Apo AI), and AMP-activated protein kinase (AMPK) phosphorylation. Additionally, EK100 also possesses analgesic and anti-inflammatory effects [7]. In spite of all the potential benefits of the plant sterol EK100, its poor aqueous solubility has hindered its wider clinical applications [8]. To date, studies examining the effect of EK100 on exercise performance, physical fatigue, and biochemical profile are lacking.

Exercise fatigue happens when exercise intensity dramatically decreases, while the athlete’s perceived effort increases. Fatigue during resistance exercise is generally attributable to neuromuscular fatigue [9] and metabolic fatigue, leading to increased acidity in the muscle fibers [10]. Metabolic fatigue early in a strength workout typically results from a depletion of phosphagen stores, while later fatigue results from impaired energy production from glycogenolysis and anaerobic glycolysis. Three categories, namely energy metabolism, oxidative stress, and inflammatory indexes, are usually assessed according to the mechanism and metabolic changes during muscle fatigue [11]. Neuromuscular fatigue is understood as the phenomenon of a decrease in athletic performance with intensive activity, causing fatigue and the termination of exercise [12].

It is well known that exercise causes an increase in reactive oxygen production (ROS), particularly in active skeletal muscle. A previous study showed that EK100 inhibits the production of tumor necrosis factor (TNF-alpha), ROS, and lipid peroxidation [7,13], thereby reducing the oxidative stress and inflammation after exhaustive exercise. Many researchers are interested in using the synergistic effects of various ingredients as a strategy in delaying fatigue and accelerating the elimination of fatigue-related metabolites. The effective dose of EK100, which is beneficial for exercise physiological adaptation and performance, as well as its safety evaluation for use, remains to be investigated prior to potential application. In this current study, we used our previously established in vivo platform to evaluate the effect of EK100 supplementation on anti-fatigue activities and exercise performance [14,15].

## 2. Results

### 2.1. Effect of EK100 Supplementation on Forelimb Grip Strength

As shown in the Figure 1A, the mean grip strength for the mice in the vehicle, EK100-1X, and EK100 groups were 117 ± 6, 120 ± 5, and 140 ± 5 g, respectively. The relative strength normalized to the individual body weight in the vehicle, EK100-1X, and EK100 groups were 318 ± 15, 315 ± 13, and 367 ± 18%, respectively (Figure 1B). The grip strength showed significant differences among the groups in absolute strength (F (2, 21) = 5.035), *p* = 0.0160) and relative strength (F (2, 21) = 3.505, *p* = 0.049). The EK100-2X supplementation groups were significantly higher than the vehicle group in regards to absolute strength (1.20- fold, *p* = 0.0220) and relative strength (1.15-fold, *p* = 0.036). Both of the grip strength measurements also showed significant EK100 dose-dependent effects (*p* = 0.0051 and *p* = 0.0480) in the trend analysis.

### 2.2. Effect of EK100 Supplementation on Exercise Performance in a Weight-loaded Swimming Test

As shown in Figure 2, time to exhaustion in the vehicle, the EK100-1X and EK100-2X groups were 2.9 ± 0.4, 5.4 ± 1.7, and 6.7 ± 2.0 min, respectively. There was no difference in the exhaustion swimming time between groups (F (2, 21) = 1.647, *p* = 0.217).

### 2.3. Effect of EK100 Supplementation on Serum Lactate, Ammonia, Glucose, Blood Urea Nitrogen (BUN), and Creatine Kinase (CK) Levels after Acute Exercise Challenge

The effects of EK100 on serum lactate, ammonia, glucose, blood urea nitrogen (BUN), and creatine kinase (CK) were evaluated after six weeks of supplementation and following a 15-min. swim test. The lactate levels in the vehicle, EK100-1X, and EK100-2X groups were 8.5 ± 0.5, 6.2 ± 0.3, and 5.8 ± 0.5 mmol/L, respectively. The EK100 groups exhibited significantly higher levels of lactate than the vehicle group (F (2, 21) = 16.072, *p* < 0.0001), corresponding to a decrease of 27.31% (*p* = 0.0002) and 31.86% (*p* < 0.0001) in the Ek100-1X and EK100-2X groups, respectively, as compared to the vehicle group (Figure 3A). This suggests that EK100 supplementation has the potential to increase the clearance or utilization of blood lactate during exercise. Figure 3B showed serum ammonia levels in the vehicle, EK100-1X, and EK100-2X groups were 176 ± 17, 109 ± 11, and 85 ± 14 μmol/L, respectively. We observed a significant difference in the ammonial levels among the groups (F (2, 21) = 11.142, *p* = 0.0010) with EK100 treatments (EK100-1X and EK100-2X) significantly lower than the vehicle group. Serum ammonia showed significant dose-dependent trends (*p* < 0.0001), suggesting that the continuous supplementation with EK100 for six weeks could mitigate ammonia accumulation during exercise.

Serum glucose levels were significantly different among groups (F (2, 21) = 3.600, *p* = 0.0450). The levels in the vehicle, EK100-1X, and EK100-2X groups were 117 ± 5, 136 ± 6, and 135 ± 6 mg/dL, respectively (Figure 3C). Trend analysis showed a dose-dependent serum glucose level increase with increased EK100 supplementation (*p* = 0.0432). Serum blood urea nitrogen (BUN) levels of the vehicle, EK100-1X, and EK100-2X groups were 24.4 ± 0.8, 21.6 ± 0.9, and 20.0 ± 1.1 mg/dL, respectively (Figure 3D). EK100 supplementation was shown to affect the BUN levels after acute exercise (F (2, 21) = 5.6930, *p* = 0.0110). In the trend analysis, the serum BUN levels decreased dose dependently with increasing EK100 dose (*p* < 0.0001). CK activities in the vehicle, EK100-1X, and EK100-2X groups were 1703 ± 136, 949 ± 261, and 759 ± 112 U/L, respectively (Figure 3E), being significantly different in each group (F (2, 21) = 7.652, *p* = 0.0030). The EK100-2X treatment group showed 55.41% (*p* = 0.0040) lower CK levels than the vehicle control. Trend analysis showed that EK100 treatment had a significant dose-dependent effect on the CK levels (*p* = 0.0001). Our findings suggest that EK100 supplementation could ameliorate skeletal muscle injury that is induced by acute exercise challenge.

### 2.4. Effect of EK100 Supplementation on Liver and Muscle Glycogen Levels

Glycogen content in the liver and muscle tissues of the mouse groups was examined (Figure 4A,B). The liver glycogen level in the vehicle, EK100-1X, and EK100-2X groups were 38.48 ± 4.45, 51.18 ± 4.46, and 57.51 ± 3.07 mg/g liver, respectively (Figure 4A). EK100 supplementation was shown to affect the glycogen content in the liver (F (2, 21) = 5.729, *p* = 0.0010). The muscle glycogen contents in the vehicle, EK100-1X, and EK100 -2X groups were 1.80 ± 0.09, 1.87 ± 0.09, and 1.85 ± 0.16 mg/g muscle, respectively (Figure 4B). Muscle glycogen levels were not significantly different between groups (F (2, 21) = 0.088, *p* = 0.9160). The EK100-1X and EK100-2X group showed significantly higher liver glycogen levels (1.33- and 1.49-fold, *p* = 0.038 and *p* = 0.009) as compared to the vehicle control group. Trend analysis revealed that EK100 treatment had a significant dose-dependent effect (*p* = 0.0001) on the liver glycogen levels.

### 2.5. Subacute Oral Toxicity Evaluation of EK100 Supplementation

Subacute toxicity evaluation to determine the safety of the EK100 supplementation was conducted following the Organization for Economic Co-operation and Development (OECD) Guideline 407. Several parameters, including behavior, diet, growth curve, organ weight, histopathology, and biochemical variables were evaluated. Table 1 summarizes the morphological data from each experimental group. The body weights (BW) for each group of mice were recorded and found to be steadily increasing throughout the experimental period (Figure 5). In addition, there was no significant difference in the initial and final BWs among the vehicle, EK100-1X, and EK100-2X groups. Trend analysis revealed that the final BW dose-dependently increased with EK-100 treatment (*p* = 0.0382). The daily intake of diet and water was no different between vehicle and EK100 groups. Additionally, no significant difference was observed in the liver, kidney, heart, lung, epididymal fat pads (EFP), brown adipose tissue (BAT), and muscle weights among the groups. We also measured the effect of EK100 on the relative tissue weight, calculated as a % by normalizing tissue weights to individual BW. There was no difference in the relative tissue weight between the groups. Our results suggest that continuous EK100 supplementation does not influence tissue weight. Histopathological examination of the major organs, including the liver, muscle, heart, kidney lung, and epididymal fat pad (EFP) did not reveal any difference between the vehicle and the EK-100 groups (Figure 6). Representative photomicrographs of each organ or tissues from each mouse group are shown.

### 2.6. Effect of EK100 Supplementation on Biochemical Markers

In order to determine the physiological adaptation status of the mice due to EK100 supplementation, we examine mice at rest for differences in the biochemical markers (Table 2). The hepatic indexes, AST and ALT, showed significant decrease in levels with EK100-1X (F(2,21) = 5.171, *p* = 0.015) and 2X supplementation (F(2,21) = 6.219, *p* = 0.008). The AST levels were lowered by 27.98% (*p* = 0.025) and 26.57% (*p* = 0.013) with the EK100-1X and EK100-2X group, respectively, as compared with the vehicle group. ALT levels were also lower by 45.19% (*p* = 0.0009) and 49.92% (*p* = 0.012) with the EK100-1X and EK100-2X groups. Trend analysis showed the AST (*p* = 0.0053) and ALT (*p* = 0.0004) levels to significantly decrease with EK100 supplementation.

The injury indexes ALP and LDH showed significant decreased levels (F(2,21) = 5.138, *p* = 0.015; F(2,21) = 6.219, *p* = 8.806, respectively) in the mice with EK100 supplementation. The ALP levels were lower by 14.95% (*p* = 0.0460) and 22.19% (*p* = 0.0130) with the EK100-1X and EK100-2X group, as compared with the vehicle group. The LDH levels were 25.28% (*p* = 0.0050) and 31.64% (*p* = 0.0020) lower with the EK100-1X and EK100-2X group, when compared with the vehicle group. In trend analysis, the ALP (*p* = 0.0011) and LDH (*p* < 0.0001) levels were significantly decreased by EK100 treatment.

Blood urea nitrogen, a fatigue-related parameter, was observed to be significantly different among the mice groups (F(2,21) = 15.349, *p* < 0.0001). EK100 supplementation significantly decreased the BUN elevation by 15.46% (*p* = 0.010, EK100-1X) and 13.15% (*p* < 0.0001 EK100-2X), showing a dose-dependent trend (*p* < 0.0001). Although the total protein levels did not significantly differ among the groups (F(2,21) = 3.095, *p* = 0.0660), trend analysis revealed a dose-dependent (*p* = 0.0061) increase by EK100 treatment.

The other indexes, including CK, energy metabolites (creatinine and glucose), blood lipid (TG and TC), and renal function (UA and albumin) were not significantly different between the groups.

## 3. Discussion

Strenuous exercise is accompanied by the increased generation of free radical products that contribute to oxidative stress [16]. Free radicals, such as reactive oxygen species (ROS) and reactive nitrogen species (RNS), are produced by normal metabolism and they are involved in various physiological processes during exercise. When there is an imbalance in the oxidative process, free radicals accumulate, leading to damage of macromolecules [17]. In a previous study [1], supplementation of mice with triterpenoid-rich extract from AC in body fruits had no effect on grip strength, but it showed an increase in swimming endurance. AC is generally marketed as capsules of crude extract and elucidating its antioxidant activity has importance towards promoting AC prescription by health care professionals [18]. It has previously been shown [7] that EK100 inhibition of MDA is correlated with the induction of antioxidant enzymes, suggesting that EK100 is a strong anti-oxidant, improving exercise performance by reducing free radical accumulation.

In addition, EK100 (10 mg/kg) has been shown to inhibit malondialdehyde (MDA), nitric oxide (NO), TNF-α, and inducible nitric oxide synthase (iNOS), and cyclooxygenase-2 (COX-2) protein expression in vivo [7]. It also exhibits a potent anti-inflammatory activity against the inflammatory response initiated by lambda-carrageenin. Previous studies suggest that the L-arginine-NO pathway mediates the anti-inflammatory effect of EK100 [19]. Improvement in exercise performance may be attributed to increased local blood flow during exercise sessions [20]. Our data suggests that different concentrations of EK100 may differently contribute to physiological activities, and that the EK100-2X (20 mg/kg) dose may be the optimal range for exercise explosiveness.

Lactate accumulation and metabolic acidosis are cellular manifestations of fatigue, with severe fatigue occurring at high blood lactate concentrations. The intense stimulation of skeletal muscle activates glycolysis, producing a high rate of lactic acid production. The acid immediately dissociates into lactate and free H^+^ [21]. The nitrogenous waste products of amino acid degradation are eliminated through the formation of urea and small amounts of ammonia. However, ammonia is toxic and associated with fatigue [22]. Blood glucose level is an important index for performance maintenance during exercise [23]. During exercise and muscle contraction, there is an increase in insulin- and contraction-stimulated glucose transport in skeletal muscle by the translocation of GLUT4 [24]. With high-intensity exercise, the levels of high-energy phosphates, ATP, and phosphocreatine (PC) decrease in the muscles, whereas inorganic phosphate (Pi), ADP, lactate, H^+^ ion, and metabolic waste, such as BUN, increase as fatigue develops [25,26,27]. These metabolic changes reflect decreased energy availability, affecting cellular regulation and contributing to fatigue development. All of these changes have been suggested as possible fatigue-inducing factors. Unusually high exercise volume can result in increased levels of creatine kinase (CK), which is indicative of muscle damage and muscle fatigue [28]. The CK levels after exercise are substantially altered within 72 h, with serum CK being an important clinical biomarker for muscle damage, such as muscular dystrophy, severe muscle breakdown, myocardial infarction, autoimmune myositides, and acute renal failure. Increased ammonia level is related to both peripheral and central fatigue during exercise. CK is a muscular damage indicator, which is released from cells into the blood stream as a result of tissue damage that is caused by excessive muscular exercise. The overproduction of lactate by muscle cells is related to excessive glycolysis resulting from high intensity exercise. Our study suggests that continuous supplementation with EK100 for six weeks could increase serum glucose levels, increase clearance of lactate and ammonia, reduce CK and BUN levels, and improve liver glycogen uptake capacity toward beneficial anti-fatigue activities.

The maintenance of blood glucose during exercise in the face of large increases in muscle glucose uptake is achieved by a concomitant increase in liver glucose output. Athletes may significantly reduce their muscle glycogen stores during exercise, which leads to muscle fatigue [29]. The increase in liver glucose output is dependent on exercise intensity and duration [30]. In our previous study [1], it was found that ethanolic extract of AC increased blood glucose, muscle glycogen and hepatic glycogen. Similar to this study, EK100 from submerged culture also increased hepatic glycogen to improve exercise performance. Additionally, our data suggests that different concentrations of EK100 may differently contribute to enhancing glycogen content, and that a 20 mg/kg dose may be the most appropriate for optimizing liver glycogen content. EK100 supplementation could help toincrease liver glycogen storage in the mice, leading to enhanced energy utilization.

Although cytotoxic studies have not been performed on EK100, we have seen that AC has a direct cytotoxicity effect against tumor cells over normal cells [3,31]. Using CCl_4_-treated rats as an experimental model, it was previously found that EK100 from *A. camphorata* has preventive properties against liver fibrosis [6]. Additionally, Liu et al., 2011 also found that AC has potent activities in suppressing the levels of ALT, AST, ALP, and total bilirubin (TB), which are elevated by alcohol intake [31]. We were able to show that EK100 protected the liver and the muscle functions, implying that EK100 prevented damage and suppressed the leakage of enzymes through cellular membranes [6]. For example, EK100 could reduce glutathione (GSH)-dependent enzymes and significantly improve the GSH/glutathione disulfide (GSSG) ratio [13]. Hepatoprotective effects may be associated with an antioxidant capacity to scavenge the reactive oxygen species. The protection of liver and muscle, as well as the anti-fatigue activity, might be due to EK100’s antioxidant properties and the inhibition of the inflammatory response.

Our data shows that EK100 did not change body composition or the daily intake of diet and water during the supplementation period. There has been no human safety study that addresses the safety concerns of EK100 in humans. In Chen et al. (2016), humans were supplemented with AC mycelium (420 mg/day, *n* = 41) for eight weeks and none of the participants experienced any adverse events during this study [32]. Here, we show that EK100 supplementation for six weeks could provide significant beneficial effects on the hepatic profile, injury index, and exercise-fatigue parameters in the mice. No tissues show gross abnormalities or changes with H&E staining at the end of the mouse study. This suggests that EK100 may be safe for potential future development with various biological efficacies. Future studies should be performed in humans to validate the safety aspects and efficacies that were established in animal models.

Above all, we found that there was no abnormal behavior observed among the groups with daily EK100 administration. Our results suggest that the continuous supplementation of EK100 supplementation does not have an impact on different organs and it may be considered to be safe for long-term supplementation at indicated doses. This study shows that EK100 has protective effects toward hepatic activity and it is able to reduce exercise injury indexes and fatigue-related parameters.

## 4. Materials and Methods

### 4.1. Preparation of Ergosta-7,9(11),22-trien-3β-ol (EK100) from Antrodia camphorata

The freeze-dried mycelium of *Antrodia camphorata* submerged whole broth was extracted with methanol. The separation and purification of EK100 is shown in Figure 7, as previously described [33].

### 4.2. Animals and Experimental Design

Male ICR mice (eight-weeks old) that were grown under specific pathogen-free conditions were purchased from BioLASCO (Yi-Lan, Taiwan). All of the mice were provided a standard laboratory diet (No. 5001; PMI Nutrition International, Brentwood, MO, USA), distilled water ad libitum, and were housed at 12-hr light/12-hr dark cycle at room temperature (22 °C ± 1 °C) and 50%–60% humidity. The Institutional Animal Care and Use Committee (IACUC) of National Taiwan Sport University (NTSU) inspected all of the animal experiments and this study conformed to the guidelines of protocol IACUC-10406 that was approved by the IACUC ethics committee. The 1X dose of EK100 used for humans is typically 7380 mg per day. The 1X mouse dose (10 mg/kg) used was converted from a human-equivalent dose (HED) based on body surface area according to the US Food and Drug Administration formula. Assuming a human weight of 60 kg, the HED for 7380 (mg)/60 (kg) = 123 × 12.3 = 10 mg/kg; the conversion coefficient of 12.3 is used to account for differences in body surface area between mice and human, as previously described [34]. In total, 24 mice were randomly assigned to three groups (eight mice/group) for daily vehicle/EK100 oral gavage for six weeks: (1) vehicle, (2) 10 mg/kg (EK100-1X), and (3) 20 mg/kg (EK100-2X). The vehicle group received the equivalent volume of deionized water based on individual body weight (BW). The mice were randomly housed in groups of four per cage.

### 4.3. Forelimb Grip Strength

A low-force testing system (Model-RX-5, Aikoh Engineering, Nagoya, Japan) was used to measure the forelimb grip strength of mice undergoing vehicle or EK100 treatment. A force transducer equipped with a metal bar (2 mm in diameter and 7.5 cm in length) for each mouse measured the amount of tensile force. The detailed procedure has been described in our previous study [35]. The test of forelimb grip strength was performed after the consecutive administration of the vehicle/EK100 for six weeks and 1 h after the last treatment dose was administered. The maximal force (in grams) that was recorded by this low-force system was recorded as the grip strength.

### 4.4. Exhaustive Swimming Test

The swim-to-exhaustion test involves loads corresponding to 5% of the mouse BW that were attached to the tails to evaluate endurance times, as previously described [36]. The swimming endurance time of each mouse was recorded from beginning to exhaustion, which is determined by observing loss of coordinated movements and failure to return to the surface within 7 s.

### 4.5. Fatigue-Associated Biochemical Indices

The effects of EK100 on serum lactate, ammonia, glucose levels, and CK activity were evaluated post-exercise, after an acute exercise challenge. One hour after the last administration, a 15-min swimming test was performed without weight loading. After the swimming exercise, the blood samples were immediately collected from the submandibular duct of mice and centrifuged at 1500× *g* and 4 °C for 10 min for serum preparation. The lactate, ammonia, glucose, BUN levels, and CK activity in the serum were determined using an autoanalyzer (Hitachi 7060, Hitachi, Tokyo, Japan). The other biochemical variables, such as aspartate aminotransferase (AST); alanine aminotransferase (ALT); alkaline phosphatase (ALP); lactate dehydrogenase (LDH); creatine kinase (CK); blood urea nitrogen (BUN); albumin; uric acid (UA); creatinine; total cholesterol (TC); triacylglycerol (TG); total protein (TP); and, glucose were measured using an autoanalyzer (Hitachi 7080) after six weeks of EK100 supplementation without exercise.

### 4.6. Tissue Glycogen Determination and Visceral Organ Weight

The stored form of glucose is glycogen, which mostly exists in the liver and muscle tissues. Liver and muscle tissues were excised after the mice were euthanized, weighed analyzed for glycogen content, as described previously [37]. The weights of the liver, kidney, heart, lung, muscle, epididymal fat pad (EFP), and brown adipose tissue (BAT) were recorded.

### 4.7. Histological Staining of Tissues

Different tissues were collected and fixed in 10% formalin after the mice were sacrificed. Following formalin fixation, the tissues were embedded in paraffin and then cut into 4-μm-thick slices for histological and pathological evaluations. Tissue sections were then stained with hematoxylin and eosin (H&E) and examined under a light microscope with a CCD camera (BX-51, Olympus, Tokyo, Japan) by a clinical pathologist.

### 4.8. Statistical Analysis

All of the data are expressed as mean ± SEM. Statistical differences among the groups were analyzed by a one-way analysis of variance (ANOVA) and the Cochran–Armitage test was used for the dose-effect trend analysis. All of the statistics were performed using the SPSS version 18.0 software (SPSS, Chicago, IL, USA), with *p*-values of < 0.05 considered to be statistically significant.

## 5. Conclusions

In the current study, we found that six-week EK100 supplementation significantly increased the explosiveness exercise. EK100 treatment showed beneficial effects on the hepatic profile, injury index, and exercise-fatigue parameters of the mice. Exercise-induced fatigue-related parameters, including lactate, ammonia, glucose, BUN, and CK levels were positively modulated by EK100 supplementation. We also found that the EK100-2X (20 mg/kg) dose may be the optimal dose for exercise performance, anti-fatigue activity, and protection of hepatic and muscle activity. Taken together, we suggest that EK100 may be a potential ergogenic aid to increase hepatic glycogen level and improve exercise performance. This directly benefits athletes by enhancing sports performance and/or maximizing training adaptations.

## Figures and Tables

**Figure 1 molecules-24-01225-f001:**
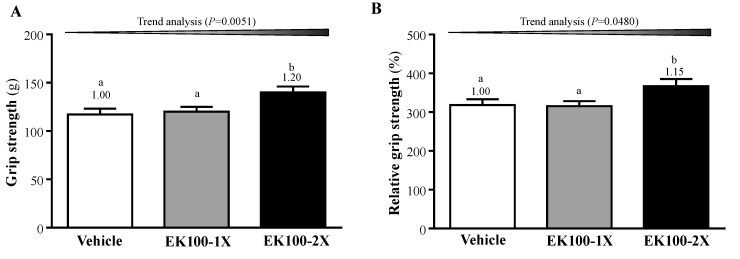
The effect of Ergosta-7,9(11),22-trien-3β-ol (EK100) supplementation on (**A**) absolute forelimb grip strength and (**B**) relative forelimb grip strength (%) to body weight (**B**). Male ICR mice were supplemented with vehicle or 10 and 20 mg/kg EK100 (1X and 2X) for six weeks before undergoing a grip strength test 1 h after the final administered dose. Data is expressed as mean ± SEM for *n* = 8 mice in each group. One-way analysis of variance (ANOVA) was used for the analysis and different letters (a, b) indicate significant difference at *p* < 0.05.

**Figure 2 molecules-24-01225-f002:**
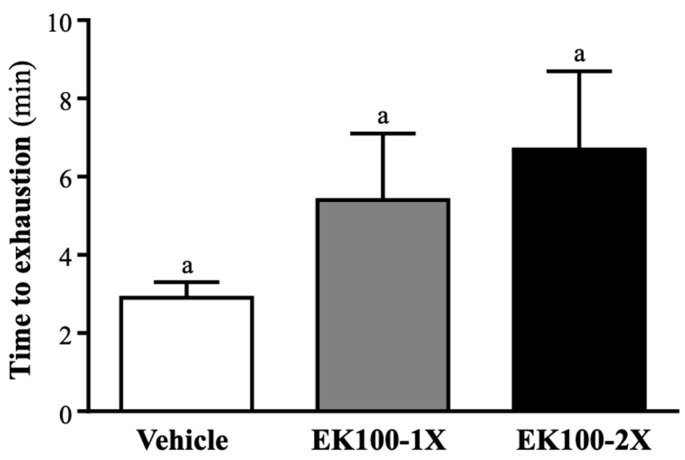
Effect of EK100 supplementation on swimming exercise performance. Mice were supplemented with vehicle, 10 and 20 mg/kg of EK100 for six weeks. One hour after the last administered dose, the mice were subjected to an exhaustive swimming exercise with a load equivalent to 5% of the mouse’s body weight attached to its tail. Data is expressed as mean ± SEM (*n* = 8 mice). Statistical analysis was done by using one-way ANOVA.

**Figure 3 molecules-24-01225-f003:**
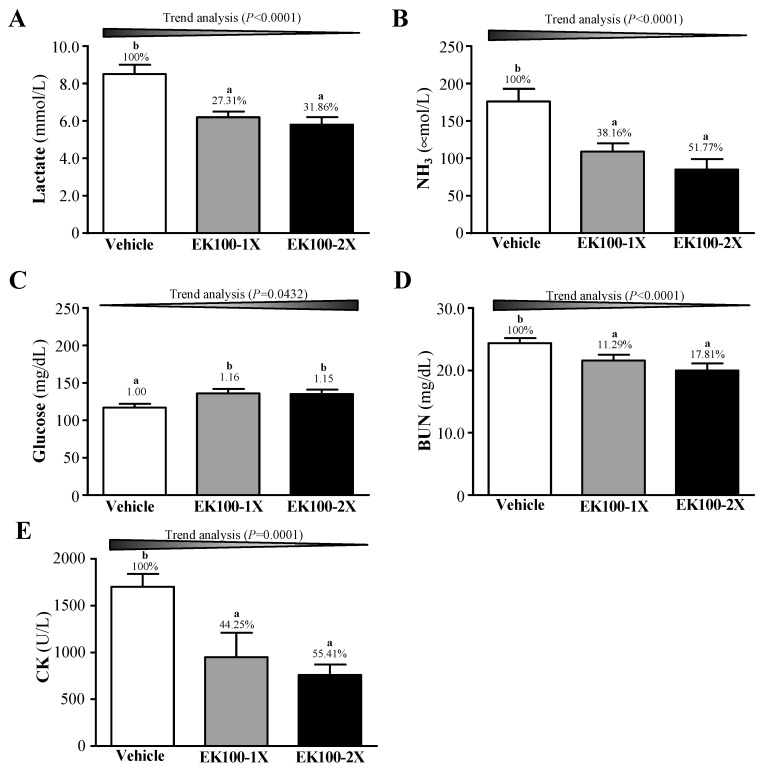
Effects of EK100 supplementation on serum (**A**) lactate, (**B**) ammonia, (**C**) glucose, (**D**) blood urea nitrogen (BUN), and (**E**) creatine kinase (CK) levels after an acute exercise challenge. Mice were supplemented with the vehicle, 10 and 20 mg/kg of EK100 for six-weeks. One hour after the last treatment dose was administered, a 15-min swimming test was performed without weight loading and serum was obtained immediately after for analysis. Data is expressed as mean ± SEM of eight mice in each group. Columns with different letters (a, b) significantly differ by one-way ANOVA (*p* < 0.05).

**Figure 4 molecules-24-01225-f004:**
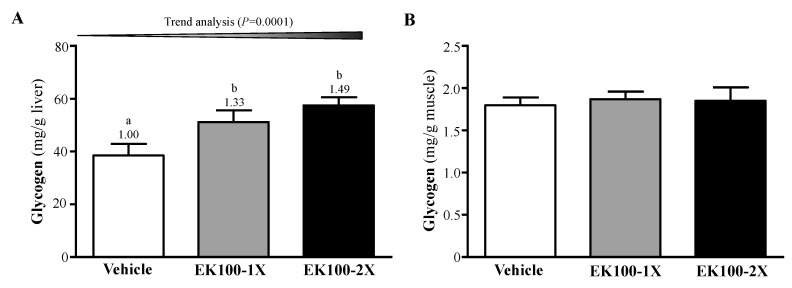
Effect of EK100 on (**A**) liver and (**B**) muscle glycogen levels at rest. Mice were supplemented with vehicle, 10 or 20 mg/kg of EK100 for six-weeks. All of the mice were sacrificed and examined for glycogen levels in muscle and liver tissues 1 h after the final treatment. Data is expressed as mean ± SEM with *n* = 8 mice in each group. Columns with different letters (a, b) significantly differ by one-way ANOVA (*p* < 0.05).

**Figure 5 molecules-24-01225-f005:**
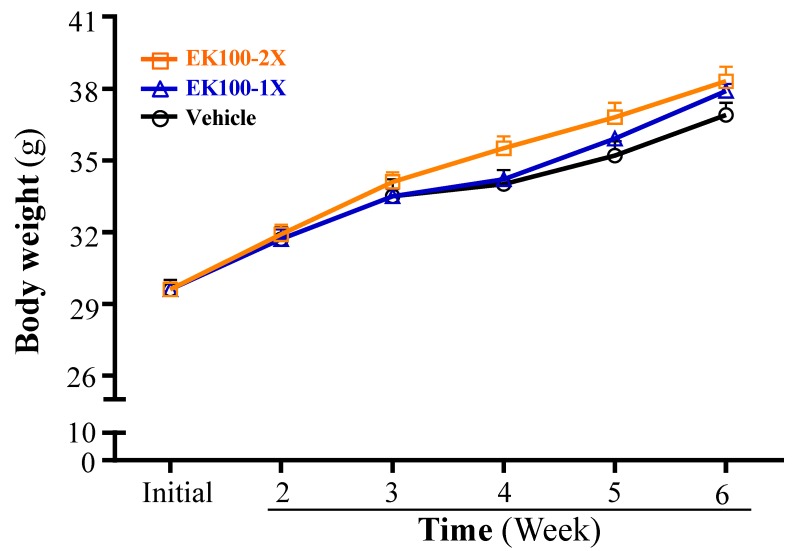
Body weight of vehicle and EK100 supplemented mice. Male ICR mice were supplemented with vehicle, 10 (EK100-1X), and 20 (EK100-2X) mg/kg EK100 for six-weeks. Data is expressed as mean ± SEM for *n* = 8 mice per group.

**Figure 6 molecules-24-01225-f006:**
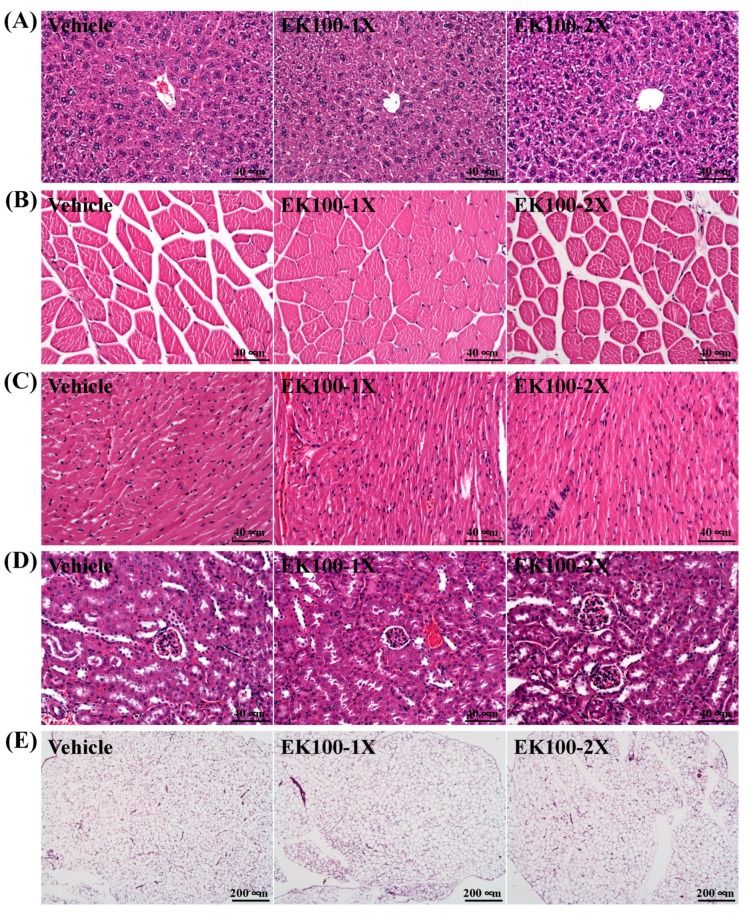
Effect of EK100 supplementation on the morphology of (**A**) liver, (**B**) skeletal muscle, (**C**) heart, (**D**) kidney, and (**E**) epididymal fat pad. Specimens were photographed with a light microscope (Olympus BX51). H&E stain, magnification: ×400, Scale bar, 10 μm.

**Figure 7 molecules-24-01225-f007:**
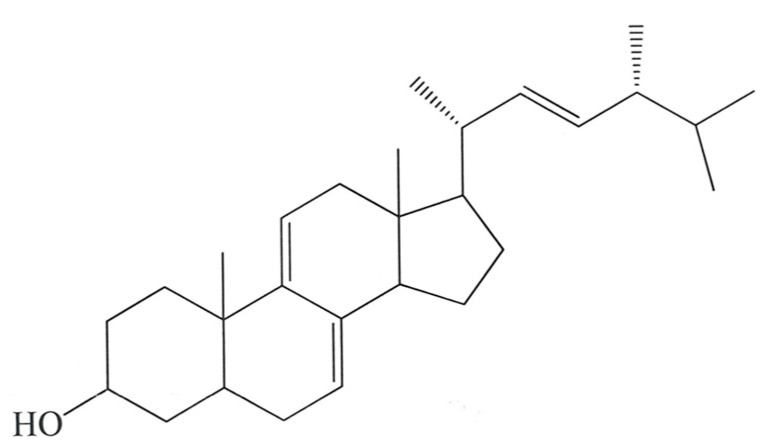
The structure of Ergosta-7,9(11),22-trien-3β-ol from *Antrodia camphorata* (EK100), molecular weight = 396.65.

**Table 1 molecules-24-01225-t001:** General characteristics of mice with EK100 supplementation.

Characteristic	Vehicle	EK100-1X	EK100-2X	Trend Analysis
Initial BW (g)	29.6 ± 0.4	29.6 ± 0.3	29.6 ± 0.3	0.7824
Final BW (g)	36.9 ± 0.5	37.9 ± 0.3	38.3 ± 0.6	0.0382
Food intake (g/day)	6.47 ± 0.15	6.38 ± 0.12	6.37 ± 0.17	0.2899
Water intake (g//day)	7.76 ± 0.23	7.71 ± 0.17	7.69 ± 0.12	0.1086
Liver (g)	2.06 ± 0.05	2.07 ± 0.03	2.09 ± 0.06	0.8734
Kidney (g)	0.58 ± 0.03	0.58 ± 0.02	0.60 ± 0.02	0.6216
EFP(g)	0.491 ± 0.03	0.525 ± 0.03	0.520 ± 0.02	0.5455
Heart (g)	0.25 ± 0.01	0.23 ± 0.01	0.25 ± 0.02	0.7737
Lung (g)	0.26 ± 0.01	0.25 ± 0.01	0.24 ± 0.01	0.3851
Muscle (g)	0.41 ± 0.01	0.41 ± 0.01	0.40 ± 0.01	0.5377
BAT (g)	0.109 ± 0.004	0.275 ± 0.102	0.211 ± 0.081	0.0586
Relative Liver weight (%)	5.58 ± 0.13	5.45 ± 0.07	5.44 ± 0.10	0.3738
Relative Kidney weight (%)	1.56 ± 0.09	1.52 ± 0.05	1.57 ± 0.05	0.7721
Relative EFP weight (%)	1.33 ± 0.08	1.39 ± 0.09	1.36 ± 0.06	0.8391
Relative Heart weight (%)	0.68 ± 0.04	0.61 ± 0.03	0.65 ± 0.04	0.4660
Relative Lung weight (%)	0.71 ± 0.04	0.65 ± 0.04	0.63 ± 0.03	0.1422
Relative Muscle weight (%)	1.12 ± 0.03	1.09 ± 0.02	1.04 ± 0.03	0.1285
Relative BAT weight	0.30 ± 0.01	0.72 ± 0.27	0.54 ± 0.20	0.0941

Data is expressed as mean ± SEM for *n* = 8 mice in each group. Statistical analysis was done by using one-way ANOVA. Muscle mass includes both gastrocnemius and soleus muscles at the back part of the lower legs. EFP: epididymal fat pad; BAT: brown adipose tissue.

**Table 2 molecules-24-01225-t002:** Biochemical analysis of mice subjected to EK100 supplementation at the end of the study.

Parameter	Vehicle	EK100-1X	EK100-2X	Trend Analysis
AST (U/L)	185 ± 13^b^	134 ± 12^a^	136 ± 14^a^	0.0053
ALT (U/L)	148 ± 24^b^	81 ± 13^a^	74 ± 7^a^	0.0004
ALP (U/L)	78 ±5^b^	66 ±3^a^	61±4^a^	0.0011
LDH (U/L)	983 ±57^b^	735± 64^a^	672 ± 43^a^	<0.0001
CK (U/L)	1744 ± 279	1385 ± 196	1360 ± 269	0.2599
BUN (mg/dL)	29.4 ± 1.7^c^	24.8 ± 0.7^b^	20.5 ± 0.7^a^	<0.0001
Albumin (g/dL)	3.5 ± 0.0	3.5 ± 0.1	3.5 ± 0.1	0.8996
UA (mg/dL)	1.43 ± 0.06	1.40 ± 0.10	1.34 ± 0.06	0.2954
TC (mg/dL)	157 ± 6	161 ± 5	148 ± 6	0.2894
TG (mg/dL)	180 ± 19	174 ± 10	146 ± 8	0.1637
Creatinine (mg/dL)	0.27 ± 0.01	0.25 ± 0.01	0.27 ± 0.02	0.9594
TP (g/dL)	4.66 ± 0.04	4.81 ± 0.03	4.84 ± 0.08	0.0061
Glucose (mg/dL)	152 ± 4	154 ± 4	167 ±7	0.0429

Data is expressed as mean ± SEM for *n* = 8 mice per group. Values in the same line with different superscripts letters (a, b, c) differ significantly (*p* < 0.05) by one-way ANOVA. AST, aspartate aminotransferase; ALT, alanine aminotransferase; ALP, alkaline phosphatase; LDH, lactate dehydrogenase; CK, creatine kinase; BUN, blood urea nitrogen; UA, uric acid; TC, total cholesterol; TG, triacylglycerol; TP, total protein.

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
