# Peer review of "The Effects of Ergosta-7,9(11),22-trien-3β-ol from Antrodia camphorata on the Biochemical Profile and Exercise Performance of Mice"

_molecules, 2019, doi:10.3390/molecules24071225_

Round 1
Reviewer 1 Report
The authors generally provided evidences that the EK100 treatment has anti-fatigue activity and protective effects of hepatic and muscle.
However, discussions on the various effects of EK100 require supplementation.
The authors cited references to the direct cytotoxicity of AC, suggesting that the EK100 has biosafety, and a description of its toxicological indicators is required.
It is necessary to discuss the possible mechanism of EK100's protection of the liver and muscle function and the anti-fatigue effect according to exercise load.
Check the expression of the results in Figure 1A on page 2.
Reviewer 2 Report
The study comes up with an interesting hypothesis, ergostatrien-3β-ol (EK100) from Antrodia camphorata (AC) a rare and unique mushroom could reduce physical fatigue, and improve exercise. The authors investigated the effect of EK100 after 6-week supplementation. EK100 treatment showed beneficial effects on the hepatic profile, injury index and exercise-fatigue parameters of the mice.
In figure 2 put the statistical symbols (a,b) on the graph.
EK100 would seem to confirm the anti-inflammatory and antioxidant activity that occurs after exercise, in the discussion improves the description of this mechanism based on your results. Supplementation of EK100 in human diet, how could it be introduced?
